# A Causal Framework for Decomposing Spurious Variations

**Drago Plecko** and **Elias Bareinboim**
Department of Computer Science
Columbia University
dp3144@columbia.edu, eb@cs.columbia.edu

## Abstract

One of the fundamental challenges found throughout the data sciences is to explain why things happen in specific ways, or through which mechanisms a certain variable $X$ exerts influences over another variable $Y$. In statistics and machine learning, significant efforts have been put into developing machinery to estimate correlations across variables efficiently. In causal inference, a large body of literature is concerned with the decomposition of causal effects under the rubric of mediation analysis. However, many variations are spurious in nature, including different phenomena throughout the applied sciences. Despite the statistical power to estimate correlations and the identification power to decompose causal effects, there is still little understanding of the properties of spurious associations and how they can be decomposed in terms of the underlying causal mechanisms. In this manuscript, we develop formal tools for decomposing spurious variations in both Markovian and Semi-Markovian models. We prove the first results that allow a non-parametric decomposition of spurious effects and provide sufficient conditions for the identification of such decompositions. The described approach has several applications, ranging from explainable and fair AI to questions in epidemiology and medicine, and we empirically demonstrate its use.

## 1 Introduction

Understanding the relationships of cause and effect is one of the core tenets of scientific inquiry and the human ability to explain why events occurred in the way they did. Hypotheses on possible causal relations in the sciences are often generated based on observing correlations in the world, after which a rigorous process using either observational or experimental data is employed to ascertain whether the observed relationships are indeed causal. One common way of articulating questions of causation is through the average treatment effect (ATE), also known as the total effect (TE), given by

$$\mathbb{E}[y \mid do(x_1)] - \mathbb{E}[y \mid do(x_0)], \tag{1}$$

where $do(\cdot)$ symbolizes the do-operator [9], and $x_0, x_1$ are two distinct values attained by the variable $X$. Instead of just quantifying the causal effect, researchers are more broadly interested in determining which causal mechanisms transmit the change from $X$ to $Y$. Such questions have received much attention and have been investigated under the rubric of causal mediation analysis [3, 12, 10, 14].

Often, however, the causal relationship may be entirely absent or account only for a part of the initially observed correlation. In these cases, the spurious (or confounded) variations between $X$ and $Y$ play a central role in explaining the phenomenon at hand. Interestingly, though, tools for decomposing spurious variations are almost entirely missing from the literature in causal inference [1].

---

[1] The only previous work which considers decompositions of spurious effects is [16]. However, this work considers the specific case of the covariance operator, and no claims are made about the general setting.

37th Conference on Neural Information Processing Systems (NeurIPS 2023).

Phenomena in which spurious variations are of central importance are abundant throughout the sciences. For instance, in medicine, the phenomenon called the *obesity paradox* signifies the counter-intuitive association of increased body fat with better survival chances in the intensive care unit (ICU) [6]. While the full explanation is still unclear, evidence in the literature suggests that the relationship is not causal [5], i.e., it is explained by spurious variations. Spurious variations also play a central role in many epidemiological investigations [13]. In occupational epidemiology, for example, the relationship of exposure to hazardous materials with cancer is confounded by other hazardous working conditions and lifestyle characteristics [4], and such spurious variations themselves may be the target of scientific inquiry. Quantities that measure such spurious variations (or a subset thereof) are called spurious effects in this paper.

Spurious variations are key in applications of fair and explainable AI as well. For instance, consider the widely recognized phenomenon in the literature known as *redlining* [15, 7], in which the location where loan applicants live may correlate with their race. Applications might be rejected based on the zip code, disproportionately affecting certain minority groups. Furthermore, in the context of criminal justice [8], the association of race with increased probability of being classified as high-risk for recidivism may in part be explained by the spurious association of race with other demographic characteristics (we take a closer look at this issue in Sec. 5). Understanding which confounders affect the relationship, and how strongly, is an important step of explaining the phenomenon, and also determining whether the underlying classifier is deemed as unfair and discriminatory.

These examples suggest that a principled approach for decomposing spurious variations may be a useful addition to the general toolkit of causal inference, and may find its applications in a wide range of settings from medicine and public health all the way to fair and explainable AI. For concreteness, in this paper we will consider the quantity

Figure 1: Exp-SE representation.

$$P(y \mid x) - P(y \mid do(x)),$$

which we will call the *experimental spurious effect* (Exp-SE, for short). This quantity, shown graphically in Fig. 1, captures the difference in variations when observing $X = x$ vs. intervening that $X = x$, which can be seen as the spurious counterpart of the total effect. Interestingly, the Exp-SE quantity is sometimes evoked in the causal inference literature, i.e.,

$$P(y \mid x) - P(y \mid do(x)) = 0 \qquad (2)$$

is known as the *zero-bias* condition [2, 9, Ch. 6]. This condition allows one to test for the existence of confounding between the variables $X$ and $Y$. A crucial observation is that, in many cases, the quantity itself may be of interest (instead of only its *null*), as it underpins the spurious variations.

Against this background, we note that tools that allow for decomposing the Exp-SE quantity currently do not exist in the literature. Our goal in this manuscript is to fill in this gap, and provide a formalism that allows for non-parametric decompositions of spurious variations. Specifically, our contributions are the following:

 (i) We introduce the notion of a partially abducted submodel (Def. 1), which underpins the inference procedure called Partial Abduction and Prediction (Alg. 2) (akin to Balke & Pearl 3-step procedure [9, Ch. 7]). Building on this new primitive, we prove the first non-parametric decomposition result for spurious effects in Markovian models (Thm. 1),

 (ii) Building on the insights coming from the new procedure, we prove the decomposition result for settings when unobserved confounding is present (Semi-Markovian models) (Thm. 3).

 (iii) We develop sufficient conditions for identification of spurious decompositions (Thm 2, 4).

## 2 Preliminaries

We use the language of structural causal models (SCMs) as our basic semantical framework [9]. A structural causal model (SCM) is a tuple $\mathcal{M} := \langle V, U, \mathcal{F}, P(u) \rangle$, where $V, U$ are sets of endogenous (observables) and exogenous (latent) variables respectively, $\mathcal{F}$ is a set of functions $f_{V_i}$, one for each $V_i \in V$, where $V_i \leftarrow f_{V_i}(\mathrm{pa}(V_i), U_{V_i})$ for some $\mathrm{pa}(V_i) \subseteq V$ and $U_{V_i} \subseteq U$. $P(u)$ is a strictly positive probability measure over $U$. Each SCM $\mathcal{M}$ is associated to a causal diagram $\mathcal{G}$ [9] over

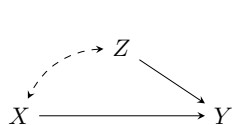
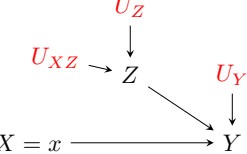

(a) Causal diagram corresponding to the SCM in Ex. 1.

(b) Extended representation of Ex. 1, latent variables in red.

(c) Diagram of Ex. 1 under $do(X = x)$ intervention.

Figure 2: Graphical representations of the SCM in Ex. 1.

the node set $V$ where $V_i \rightarrow V_j$ if $V_i$ is an argument of $f_{V_j}$, and $V_i \leftarrow\!-\!\rightarrow V_j$ if the corresponding $U_{V_i}, U_{V_j}$ are not independent [2]. A model with no bidirected edges is called *Markovian*, while a model with bidirected edges is called Semi-Markovian. An instantiation of the exogenous variables $U = u$ is called a *unit*. By $Y_x(u)$ we denote the potential response of $Y$ when setting $X = x$ for the unit $u$, which is the solution for $Y(u)$ to the set of equations obtained by evaluating the unit $u$ in the submodel $\mathcal{M}_x$, in which all equations in $\mathcal{F}$ associated with $X$ are replaced by $X = x$. In a slight abuse of notation, we also replace $Y = y$ with just $y$ whenever the former is clear from the context. We next introduce an important inferential procedure for solving different tasks in causal inference.

## 2.1 Abduction, Action and Prediction

The steps of the *abduction-action-prediction* method can be summarized as follows:

**Algorithm 1** (Abduction, Action and Prediction [9]). *Given an SCM $\langle \mathcal{F}, P(u) \rangle$, the conditional probability $P(Y_C \mid E = e)$ of a counterfactual sentence "if it were $C$ then $Y$", upon observing the evidence $E = e$, can be evaluated using the following three steps:*

*(i)* **Abduction** *– update $P(u)$ by the evidence $e$ to obtain $P(u \mid e)$,*

*(ii)* **Action** *– modify $\mathcal{F}$ by the action $do(C)$, where $C$ is an antecedent of $Y$, to obtain $\mathcal{F}_C$,*

*(iii)* **Prediction** *– use the model $\langle \mathcal{F}_C, P(u \mid e) \rangle$ to compute the probability of $Y_C$.*

In the first step, the probabilities of the exogenous variables $U$ are updated according to the observed evidence $E = e$. Next, the model $\mathcal{M}$ is modified to a submodel $\mathcal{M}_C$. The action step allows one to consider queries related to interventions or imaginative, counterfactual operations. In the final step, the updated model $\langle \mathcal{F}_C, P(u \mid e) \rangle$ is used to compute the conditional probability $P(y_C \mid e)$. There are two important special cases of the procedure. Whenever the action step is empty, the procedure handles queries in the first, associational layer of the Pearl's Causal Hierarchy (PCH, [2]). Whenever the abduction step is empty, but the action step is not, the procedure handles *interventional* queries in the second layer of the PCH. The combination of the two steps, more generally, allows one to consider queries in all layers of the PCH, including the third, *counterfactual* layer. In the following example, we look at the usage of the procedure on some queries.

**Example 1** (Abduction, Action, Prediction). *Consider the following SCM:*

$$\mathcal{F} : \begin{cases} X \leftarrow f_X(U_X, U_{XZ}) & (3) \\ Z \leftarrow f_Z(U_Z, U_{XZ}) & (4) \\ Y \leftarrow f_Y(X, Z, U_Y), & (5) \end{cases}$$

*with $P(U_X, U_{XZ}, U_Z, U_Y)$ the distribution over the exogenous variables. The causal diagram of the model is shown in Fig. 2a, with an explicit representation of the exogenous variables in Fig. 2b.*

*We are first interested in the query $P(y \mid x)$ in the given model. Based on the abduction-prediction procedure, we can simply compute that:*

$$P(y \mid x) = \sum_u \mathbb{1}(Y(u) = y)P(u \mid x) = \sum_u \mathbb{1}(Y(u) = y)P(u_z, u_y)P(u_x, u_{xz} \mid x). \quad (6)$$

*where the first step follows from the definition of the observational distribution, and the second step follows from noting the independence $U_Z, U_Y \perp\!\!\!\perp U_X, U_{XZ}, X$. In the abduction step, we can compute the probabilities $P(u_x, u_{xz} \mid x)$. In the prediction step, query $P(y \mid x)$ is computed based on Eq. 6.*

*Based on the procedure, we can also compute the query $P(y_x)$ (see Fig. 2c):*

$$P(y_x) = \sum_u \mathbb{1}(Y_x(u) = y)P(u) = \sum_u \mathbb{1}(Y(x, u_{xz}, u_z, u_y) = y)P(u). \tag{7}$$

*where the first step follows from the definition of an interventional distribution, and the second step follows from noting that $Y_x$ does not depend on $u_x$. In this case, the abduction step is void, since we are not considering any specific evidence $E = e$. The value of $Y(x, u_{xz}, u_z, u_y)$ can be computed from the submodel $\mathcal{M}_x$. Finally, using Eq. 7 we can perform the prediction step. We remark that*

$$\mathbb{1}(Y(x, u_{xz}, u_z, u_y) = y) = \sum_{u_x} \mathbb{1}(Y(u_x, u_{xz}, u_z, u_y) = y)P(u_x \mid x, u_{xz}, u_z, u_y), \tag{8}$$

*by the law of total probability and noting that $X$ is a deterministic function of $u_x, u_{xz}$. Thus, $P(y_x)$ also admits an alternative representation*

$$P(y_x) = \sum_u \mathbb{1}(Y(u_x, u_{xz}, u_z, u_y) = y)P(u_x \mid x, u_{xz}, u_z, u_y)P(u_{xz}, u_z, u_y) \tag{9}$$

$$= \sum_u \mathbb{1}(Y(u) = y)P(u_x \mid x, u_{xz})P(u_{xz}, u_z, u_y), \tag{10}$$

*where Eq. 10 follows from using the independencies among $U$ and $X$ in the graph in Fig. 2b. We revisit the representation in Eq. 10 in Ex. 2.*

## 3 Foundations of Decomposing Spurious Variations

After getting familiar with the abduction-action-prediction procedure, our next task is to introduce a new procedure that allows us to decompose spurious effects. First, we define the concept of a *partially abducted submodel*:

**Definition 1** (Partially Abducted Submodel). *Let $U_1, U_2 \subseteq U$ be a partition of the exogenous variables. Let the partially abducted (PA, for short) submodel with respect to the exogenous variables $U_1$ and evidence $E = e$ be defined as:*

$$\mathcal{M}^{U_1, E=e} := \langle \mathcal{F}, P(u_1)P(u_2 \mid u_1, E) \rangle. \tag{11}$$

In words, in the PA submodel, the typically obtained posterior distribution $P(u \mid e)$ is replaced by the distribution $P(u_2 \mid u_1, e)$. Effectively, the exogenous variables $U_1$ are *not updated according to evidence*. The main motivation for introducing the PA model is that spurious variations arise whenever we are comparing units of the population that are different, a realization dating back to Pearson in the 19th century [11]. To give a formal discussion on what became known as *Pearson's shock*, consider two sets of differing evidence $E = e$ and $E = e'$. After performing the abduction step, the variations between posterior distributions $P(u \mid e)$ and $P(u \mid e')$ will be explained by *all the exogenous variables that precede the evidence $E$*. In a PA submodel, however, the posterior distribution $P(u_1)P(u_2 \mid u_1, e)$ will differ from $P(u_1)P(u_2 \mid u_1, e')$ only in variables that are in $U_2$, while the variables in $U_1$ will induce no spurious variations. Note that if $U_1 = U$, then the PA submodel will introduce no spurious variations, a point to which we return in the sequel.

We now demonstrate how the definition of a PA submodel can be used to obtain partially abducted conditional probabilities:

**Proposition 1** (PA Conditional Probabilities). *Let $P(Y = y \mid E = e^{U_1})$ denote the conditional probability of the event $Y = y$ conditional on evidence $E = e$, defined as the probability of $Y = y$ in the PA submodel $\mathcal{M}^{U_1, E=e}$ (i.e., the exogenous variables $U_1$ are not updated according to the evidence). Then, we have that:*

$$P(Y = y \mid E = e^{U_1}) = \sum_{u_1} P(U_1 = u_1)P(Y = y \mid E = e, U_1 = u_1). \tag{12}$$

### 3.1 Partial Abduction and Prediction

Based on the notion of a PA submodel, we can introduce the partial-abduction and prediction procedure:

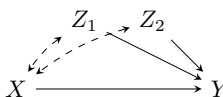

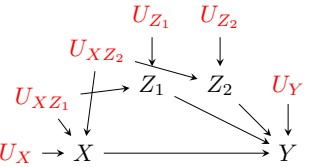

(a) Causal diagram corresponding to the SCM in Ex. 3.

(b) Extended graphical representation of the SCM in Ex. 3, latent variables in red.

Figure 3: Graphical representations of the SCM in Ex. 1.

**Algorithm 2** (Partial Abduction and Prediction). *Given an SCM $\langle \mathcal{F}, P(u) \rangle$, the conditional probability $P(Y = y \mid E = e^{U_1})$ of an event $Y = y$ upon observing the evidence $e$, in a world where variables $U_1$ are unresponsive to evidence, can be evaluated using the following two steps:*

    *(i) **Partial Abduction** – update $P(u)$ by the evidence $e$ to obtain $P(u_1)P(u_2 \mid u_1, e)$, where $(u_1, u_2)$ is a partition of the exogenous variables $u$,*

    *(ii) **Prediction** – use the model $\langle \mathcal{F}, P(u_1)P(u_2 \mid u_1, e) \rangle$ to compute the probability of $Y = y$.*

In the first step of the algorithm, we only perform *partial abduction*. The exogenous variables $U_2$ are updated according to the available evidence $E = e$, while the variables $U_1$ retain their original distribution $P(u_1)$ and remain unresponsive to evidence. This procedure allows us to consider queries in which only a subset of the exogenous variables respond to the available evidence. We next explain what kind of queries fall within this scope, beginning with an example:

**Example 2** (Partial Abduction and Prediction). *Consider the model in Eq. 3-5. We are interested in computing the query:*

$$P(y \mid x^{U_{xz}, U_z}) = \sum_u \mathbb{1}(Y(u) = y)P(u_{xz}, u_z)P(u_x, u_y \mid u_{xz}, u_x, x) \tag{13}$$

$$= \sum_u \mathbb{1}(Y(u) = y)P(u_{xz}, u_z)P(u_y)P(u_x \mid u_{xz}, u_x, x) \tag{14}$$

$$= \sum_u \mathbb{1}(Y(u) = y)P(u_{xz}, u_z, u_y)P(u_x \mid u_{xz}, u_x, x), \tag{15}$$

*where the first step follows from Prop. 1, and the remaining steps from conditional independencies between the $U$ variables and $X$. Crucially, the query yields the same expression as in Eq. 10 that we obtained for $P(y_x)$ in Ex. 1. Therefore, the conditional probability $P(y \mid x^{U_{xz}, U_z})$ in a world where $U_{XZ}, U_Z$ are unresponsive to evidence is equal to the interventional probability $P(y_x)$.*

As the example illustrates, we have managed to find another procedure that mimics the behavior of the interventional ($do(X = x)$) operator in the given example. Interestingly, however, in this procedure, we have not made use of the submodel $\mathcal{M}_x$ that was used in the abduction-action-prediction procedure. We next introduce an additional example that shows how the new procedure allows one to decompose spurious variations in causal models:

**Example 3** (Spurious Decomposition). *Consider an SCM compatible with the graphical representation in Fig. 3b (with exogenous variables $U$ shown explicitly in red), and the corresponding Semi-Markovian causal diagram in Fig. 3a. We note that, based on the partial abduction-prediction procedure, the following two equalities hold:*

$$P(y \mid x) = P(y \mid x^{\emptyset}) \tag{16}$$

$$P(y_x) = P(y \mid x^{U_{xz_1}, U_{xz_2}}), \tag{17}$$

*which shows that*

$$\textit{Exp-SE}_x(y) = P(y \mid x^{\emptyset}) - P(y \mid x^{U_{xz_1}, U_{xz_2}}). \tag{18}$$

*The experimental spurious effect can be written as a difference of conditional probabilities $y \mid x$ in a world where all variables $U$ are responsive to evidence vs. a world in which $U_{XZ_1}, U_{XZ_2}$ are*

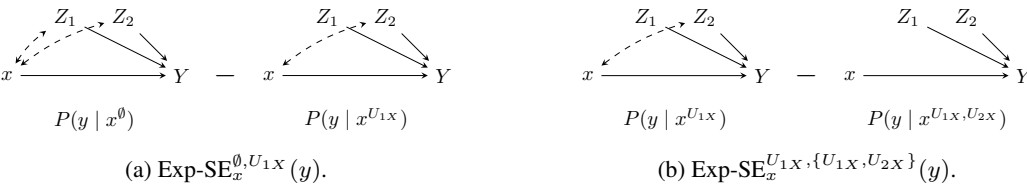

(a) Exp-SE$_x^{\emptyset, U_1X}(y)$.  (b) Exp-SE$_x^{U_1X, \{U_1X, U_2X\}}(y)$.

Figure 4: Graphical representation of how the Exp-SE effect is decomposed in Ex. 3.

| Procedure | SCM | Queries |
|---|---|---|
| Abduction-Prediction | $\langle \mathcal{F}, P(u \mid E) \rangle$ | Layer 1 |
| Action-Prediction | $\langle \mathcal{F}_x, P(u) \rangle$ | Layer 2 |
| Abduction-Action-Prediction | $\langle \mathcal{F}_x, P(u \mid E) \rangle$ | Layers 1, 2, 3 |
| Partial Abduction-Prediction | $\langle \mathcal{F}, P(u_1)P(u_2 \mid E) \rangle$ | Layers 1, 2, 3 |

Table 1: Summary of the different procedures and the corresponding probabilistic causal models.

*unresponsive to evidence. Furthermore, we can also consider a refinement that decomposes the effect*

$$Exp\text{-}SE_x(y) = \underbrace{P(y \mid x^{\emptyset}) - P(y \mid x^{U_{xz_1}})}_{\text{variations of } U_{xz_1}} + \underbrace{P(y \mid x^{U_{xz_1}}) - P(y \mid x^{U_{xz_1}, U_{xz_2}})}_{\text{variations of } U_{xz_2}}, \quad (19)$$

*allowing for an additive, non-parametric decomposition of the experimental spurious effect.*

The first term in Eq. 19, shown in Fig. 8a, encompasses spurious variations explained by the variable $U_{XZ_1}$. The second term, in Fig. 4b, encompasses spurious variations explained by $U_{XZ_2}$.

For an overview, in Tab. 1 we summarize the different inferential procedures discussed so far, indicating the structural causal models associated with them.

## 4 Non-parametric Spurious Decompositions

We now move on to deriving general decomposition results for the spurious effects. Before doing so, we first derive a new decomposition result for the TV measure, not yet appearing in the literature (due to space constraints, all proofs are given in Appendix A):

**Proposition 2.** *Define the total variation (TV) measure as* $TV_{x_0, x_1}(y) = P(y \mid x_1) - P(y \mid x_0)$, *and the total effect TE as* $TE_{x_0, x_1}(y) = P(y_{x_1}) - P(y_{x_0})$. *The total variation measure can be decomposed as:*

$$TV_{x_0, x_1}(y) = TE_{x_0, x_1}(y) + (Exp\text{-}SE_{x_1}(y) - Exp\text{-}SE_{x_0}(y)). \quad (20)$$

The above result clearly separates out the causal variations (measured by the TE) and the spurious variations (measured by Exp-SE terms) within the TV measure. The seminal result from [10] can be used to further decompose the TE measure. In the sequel, we show how the Exp-SE terms can be further decomposed, thereby reaching a full non-parametric decomposition of the TV measure.

### 4.1 Spurious Decompositions for the Markovian case

When using the definition of a PA submodel, the common variations between $X, Y$ can be attributed to (or explained by) the unobserved confounders $U_1, \ldots, U_k$. In order to do so, we first define the notion of an experimental spurious effect for a set of latent variables:

**Definition 2** (Spurious effects for Markovian models). *Let $\mathcal{M}$ be a Markovian model. Let $Z_1, \ldots, Z_k$ be the confounders between variables $X$ and $Y$ sorted in any valid topological order, and denote the corresponding exogenous variables as $U_1, \ldots, U_k$, respectively. Let $Z_{[i]} = \{Z_1, \ldots, Z_i\}$ and*

$Z_{-[i]} = \{Z_{i+1}, \ldots, Z_k\}$. $U_{[i]}$ and $U_{-[i]}$ are defined analogously. Define the experimental spurious effect associated with variable $U_{i+1}$ as

$$Exp\text{-}SE_x^{U_{[i]}, U_{[i+1]}}(y) = P(y \mid x^{U_{[i]}}) - P(y \mid x^{U_{[i+1]}}). \tag{21}$$

The intuition behind the quantity $Exp\text{-}SE_x^{U_{[i]}, U_{[i+1]}}(y)$ can be explained as follows. The quantity $P(y \mid x^{U_{[i]}})$ captures all the variations in $Y$ induced by observing that $X = x$ apart from those explained by the latent variables $U_1, \ldots, U_i$, which are fixed a priori and not updated. Similarly, the quantity $P(y \mid x^{U_{[i+1]}})$ captures the variations in $Y$ induced by observing that $X = x$, apart from those explained by $U_1, \ldots, U_i, U_{i+1}$. Therefore, taking the difference of the two quantities measures the variation in $Y$ induced by observing that $X = x$ that is explained by the latent variable $U_{i+1}$.

Based on this definition, we can derive the first key non-parametric decomposition of the experimental spurious effect that allows the attribution of the spurious variations to the latent variables $U_i$:

**Theorem 1** (Latent spurious decomposition for Markovian models). *The experimental spurious effect $Exp\text{-}SE_x(y)$ can be decomposed into latent variable-specific contributions as follows:*

$$Exp\text{-}SE_x(y) = \sum_{i=0}^{k-1} Exp\text{-}SE_x^{U_{[i]}, U_{[i+1]}}(y) = \sum_{i=0}^{k-1} P(y \mid x^{U_{[i]}}) - P(y \mid x^{U_{[i+1]}}). \tag{22}$$

An illustrative example of applying the theorem is shown in Appendix B.1. Thm. 1 allows one to attribute spurious variations to latent variables influencing both $X$ and $Y$. The key question is when such an attribution, as shown in Eq. 22, can be computed from observational data in practice (known as an *identifiability* problem [9]). In fact, when variables are added to the PA submodel in topological order, the attribution of variations to the latents $U_i$ is identifiable, as we prove next:

**Theorem 2** (Spurious decomposition identification in topological ordering). *The quantity $P(y \mid x^{U_{[i]}})$ can be computed from observational data using the expression*

$$P(y \mid x^{U_{[i]}}) = \sum_z P(y \mid z, x) P(z_{-[i]} \mid z_{[i]}, x) P(z_{[i]}), \tag{23}$$

*rendering each term of decomposition in Eq. 22 identifiable from the observational distribution $P(v)$.*

We discuss in Appendix B.2 why a decomposition that does not follow a topological order of the variables $U_i$ is not identifiable.

## 4.2 Spurious Decompositions in Semi-Markovian Models

In the Markovian case, considered until now, there was a one-to-one correspondence between the observed confounders $Z_i$ and their latent variables $U_i$. This, however, is no longer the case in Semi-Markovian models. In particular, it can happen that there exist exogenous variables $U_j$ that induce common variations between $X, Y$, but affect more than one confounder $Z_i$. We are interested in $U_j \subseteq U$ that have causal (directed) paths to both $X, Y$, described by the following definition:

**Definition 3** (Trek). *Let $\mathcal{M}$ be an SCM corresponding to a Semi-Markovian model. Let $\mathcal{G}$ be the causal diagram of $\mathcal{M}$. A trek $\tau$ in $\mathcal{G}$ (from $X$ to $Y$) is an ordered pair of causal paths $(g_l, g_r)$ with a common exogenous source $U_i \in U$. That is, $g_l$ is a causal path $U_i \to \cdots \to X$ and $g_r$ is a causal path $U_i \to \cdots \to Y$. The common source $U_i$ is called the top of the trek (ToT for short), denoted $top(g_l, g_r)$. A trek is called spurious if $g_r$ is a causal path from $U_i$ to $Y$ that is not intercepted by $X$.*

When decomposing spurious effects, we are in fact interested in all the exogenous variables $U_i$ that lie on top of a spurious trek between $X$ and $Y$. It is precisely these exogenous variables that induce common variations between $X$ and $Y$. Using any subset of the variables that are top of spurious treks, we define a set-specific notion of a spurious effect:

**Definition 4** (Exogenous set-specific spurious effect). *Let $U_{sToT} \subseteq U$ be the subset of exogenous variables that lie on top of a spurious trek between $X$ and $Y$. Suppose $A, B \subseteq U_{sToT}$ are two nested subsets of $U_{sToT}$, that is $A \subseteq B$. We then define the exogenous experimental spurious effect with respect to sets $A, B$ as*

$$Exp\text{-}SE_x^{A,B}(y) = P(y \mid x^A) - P(y \mid x^B). \tag{24}$$

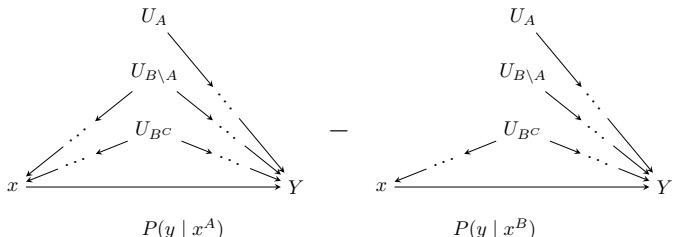

Figure 5: Quantity Exp-SE$_x^{A,B}(y)$ as a graphical contrast. Dots $\cdots$ indicate arbitrary observed confounders along the indicated pathway.

The above definition is analogous to Def. 2, but we are now fixing different subsets of the tops of spurious treks. Def. 2 supports partial abduction of exogenous variables that are not on top of a spurious trek, but we are seldom interested in these since they do not induce covariations of $X, Y$. The quantity Exp-SE$_x^{A,B}(y)$ is presented as a graphical contrast in Fig. 5. In particular, the set of tops of spurious treks $U_{sToT}$ is partitioned into three parts $(U_A, U_{B \setminus A}, U_{B^C})$. The causal diagram in the figure is informal, and the dots $(\cdots)$ represent arbitrary possible observed confounders that lie along indicated pathways. On the l.h.s. of the figure, the set $U_A$ does not respond to the conditioning $X = x$, whereas $U_{B \setminus A}, U_{B^C}$ do. This is contrasted with the r.h.s., in which neither $U_A$ nor $U_{B \setminus A}$ respond to $X = x$, whereas $U_{B^C}$ still does respond to the $X = x$ conditioning. The described contrast thus captures the spurious effect explained by the tops of spurious treks in $U_{B \setminus A}$.

Analogous to Thm. 1, we next state a variable-specific decomposition of the spurious effect, which is now with respect to exogenous variables that are top of spurious treks:

**Theorem 3** (Semi-Markovian spurious decomposition). *Let $U_{sToT} = \{U_1, \ldots, U_m\} \subseteq U$ be the subset of exogenous variables that lie on top of a spurious trek between $X$ and $Y$. Let $U_{[i]}$ denote the variables $U_1, \ldots, U_i$ ($U_{[0]}$ denotes the empty set $\emptyset$). The experimental spurious effect Exp-SE$_x(y)$ can be decomposed into variable-specific contributions as follows:*

$$Exp\text{-}SE_x(y) = \sum_{i=0}^{m-1} Exp\text{-}SE_x^{U_{[i]}, U_{[i+1]}}(y) = \sum_{i=0}^{k-1} P(y \mid x^{U_{[i]}}) - P(y \mid x^{U_{[i+1]}}). \tag{25}$$

An example demonstrating the Semi-Markovian decomposition is given in Appendix B.3. We next discuss the question of identification. We begin by discussing how to annotate the exogenous variables given a Semi-Markovian causal diagram:

**Definition 5** (Top of trek from the causal diagram). *Let $\mathcal{M}$ be a Semi-Markovian model and let $\mathcal{G}$ be the associated causal diagram. A set of nodes fully connected with bidirected edges is called a clique. A maximal clique $C_i$ is such that there is no clique $C_i'$ such that $C_i \subsetneq C_i'$. The set of variables $U_{sToT}$ can be constructed from the causal diagram in the following way:*

(I) *initialize $U_{sToT} = \emptyset$,*

(II) *for each maximal clique $C_i$, consider the associated exogenous variable $U_{C_i}$ pointing to each node in the clique; if there exists a spurious trek between $X$ and $Y$ with a top in $U_{C_i}$, add $U_{C_i}$ to $U_{sToT}$.*

After defining the explicit construction of the set $U_{sToT}$, we define the notion of the anchor set:

**Definition 6** (Anchor Set). *Let $U_1, \ldots U_l \subseteq U$ be a subset of the exogenous variables. We define the anchor set $AS(U_1, \ldots, U_l)$ of $(U_1, \ldots, U_l)$ as the subset of observables $V$ that are directly influenced by any of the $U_i$s,*

$$AS(U_1, \ldots, U_l) = \bigcup_{i=1}^{l} \text{ch}(U_i). \tag{26}$$

Another important definition is that of anchor set exogenous ancestral closure:

|  SCM $\mathcal{M}$  |  Causal Diagram $\mathcal{G}$  |

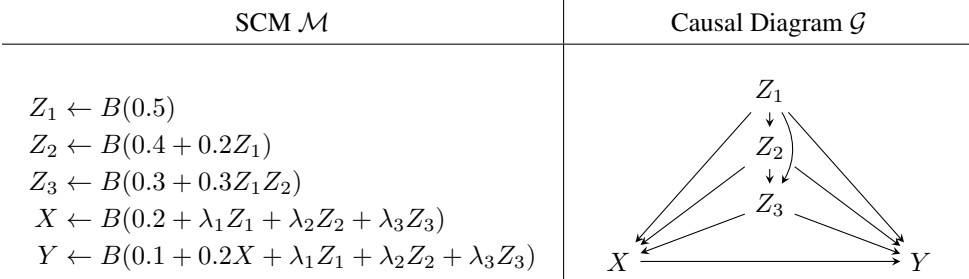

$$Z_1 \leftarrow B(0.5)$$
$$Z_2 \leftarrow B(0.4 + 0.2Z_1)$$
$$Z_3 \leftarrow B(0.3 + 0.3Z_1 Z_2)$$
$$X \leftarrow B(0.2 + \lambda_1 Z_1 + \lambda_2 Z_2 + \lambda_3 Z_3)$$
$$Y \leftarrow B(0.1 + 0.2X + \lambda_1 Z_1 + \lambda_2 Z_2 + \lambda_3 Z_3)$$

Table 2: SCM and causal diagram for the Synthetic A example.

**Definition 7** (Anchor Set Exogenous Ancestral Closure). *Let $U_s \subseteq U$ be a subset of the exogenous variables. Let $AS(U_s)$ denote the anchor set of $U_s$, and let $\mathrm{an}^{ex}(AS(U_s))$ denote all exogenous variables that have a causal path to any variable in $AS(U_s)$. $U_s$ is said to satisfy anchor set exogenous ancestral closure (ASEAC) if*

$$U_s = \mathrm{an}^{ex}(AS(U_s)). \tag{27}$$

Based on the above, we provide a sufficient condition for identification in the Semi-Markovian case:

**Theorem 4** (ID of variable spurious effects in Semi-Markovian models). *Let $U_s \subseteq U_{sToT}$. The quantity $P(y \mid x^{U_s})$ is identifiable from observational data $P(V)$ if the following hold:*

*(i) $X \notin AS(U_s)$, $Y \notin AS(U_s)$*

*(ii) $U_s$ satisfies anchor set exogenous ancestral closure, $U_s = \mathrm{an}^{ex}(AS(U_s))$.*

Some instructive examples grounding Defs. 5-7 and Thm. 4 can be found in Appendix B.4. In words, the conditional expectation of $Y$ given $X$ in the partially abducted submodel w.r.t. a set $U_s$ is identifiable whenever (i) neither $X$ nor $Y$ are elements of the anchor set of $U_s$ and (ii) the set $U_s$ satisfies the anchor set exogenous ancestral closure. Thm. 4 provides a sufficient, but not a necessary condition for identification. An additional discussion of the conditions is given in Appendix C. We hope to address in future work an algorithmic way for identifying spurious effects in full generality.

## 5 Experiments

We now apply our framework to a synthetic example (called Synthetic A) with a known ground truth, summarized in Tab. 2 where the SCM $\mathcal{M}$ and the causal diagram $\mathcal{G}$ are given. The source code for the experiment can be found in our repository. For this example, we set the parameters $\lambda_1 = \lambda_2 = \lambda_3 = 0.2$. We then vary each parameter $\lambda_i \in [0, 0.2]$ (while keeping the other two parameters fixed), which changes the value of the effect associated with latent variable $U_i$. The effects associated with each $U_i, i \in \{1, 2, 3\}$ are computed based on the decomposition in Thm. 1:

$$\text{Exp-SE}_x^{U_1}(y) := \text{Exp-SE}_x^{\emptyset, U_1}(y) = P(y \mid x^{\emptyset}) - P(y \mid x^{U_1}) \tag{28}$$

$$\text{Exp-SE}_x^{U_2}(y) := \text{Exp-SE}_x^{U_1, \{U_1, U_2\}}(y) = P(y \mid x^{U_1}) - P(y \mid x^{U_1, U_2}) \tag{29}$$

$$\text{Exp-SE}_x^{U_3}(y) := \text{Exp-SE}_x^{\{U_1, U_2\}, \{U_1, U_2, U_3\}}(y) = P(y \mid x^{U_1, U_2}) - P(y \mid x^{U_1, U_2, U_3}). \tag{30}$$

The key task is to compute the ground truth values of $P(y \mid x^{U_{[i]}})$ for different values of $i$. According to Def. 1, we want to obtain the conditional distribution of $Y$ given $X = x$ but subject to not updating $U_{[i]}$ according to the evidence $X = x$. Based on the true SCM, this can be done efficiently using rejection sampling as follows:

(1) Take $N$ samples from the SCM $\mathcal{M}$ in Tab. 2,

(2) For all samples $k \in \{1, \ldots, N\}$ with $u^{(k)}$ such that

$$X(u^{(k)}) \neq x, \tag{31}$$

re-sample the part of the unit $u^{(k)}$ that is not included in $U_{[i]}$ (e.g., if $U_{[i]} = \{U_1, U_2\}$, latent $u_1^{(k)}, u_2^{(k)}$ are not re-sampled but $u_3^{(k)}$ is) and replace $u^{(k)}$ with this new sample,

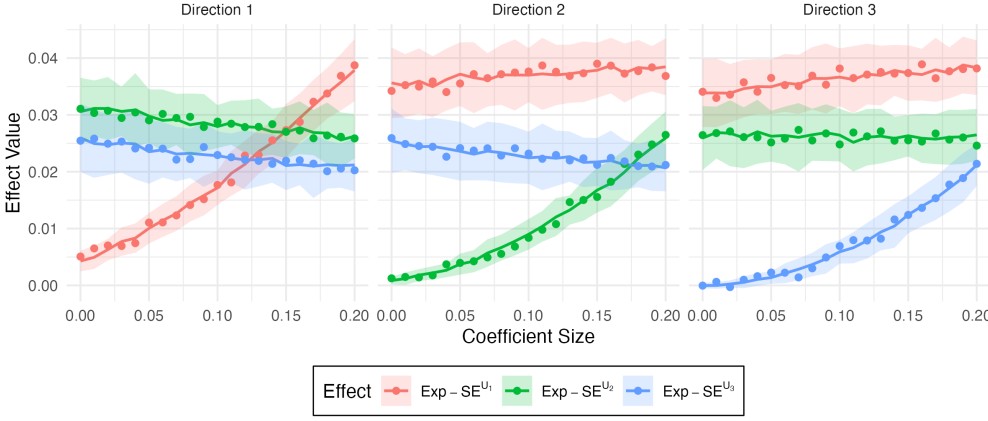

Figure 6: Experimental results on the Synthetic A example. Lines indicate the estimated values, dots the ground truth obtained from the SCM using rejection sampling, and the 95% confidence intervals are indicated with color. As expected, increasing the $\lambda_i$ coefficient increases the spurious effect associated with the latent variable $U_i$.

(3) Evaluate the mechanisms $\mathcal{F}$ of $\mathcal{M}$ for all units $u^{(k)}$,

(4) If there exists a sample $k$ with $X(u^{(k)}) \neq x$ go back to Step (2),

(5) Return the mean of the $Y$ variables $\frac{1}{N} \sum_{k=1}^{N} Y^{(k)}$.

Notice that the described procedure gives us samples from the distribution $P(y \mid x^{U_{[i]}})$. The values of $U_{[i]}$ are sampled only once and are not updated after the initial sampling. Other values in $U$, however, are sampled anew until their values are such that they are compatible with the evidence $X = x$. Therefore, the procedure guarantees that $U_{[i]}$ do not respond to the evidence, whereas the complement of $U_{[i]}$ does, allowing us to compute $P(y \mid x^{U_{[i]}})$ and in turn the expressions in Eqs. 28-30. The effects are also estimated from observational data based on the identification expressions in Thm. 2. Fig. 6 demonstrates that the SCM-based ground truth matches the estimates based on Thm. 2.

## 6 Conclusions

In this paper, we introduced a general toolkit for decomposing spurious variations in causal models. In particular, we introduced a new primitive called *partially abducted submodel* (Def. 1), together with the procedure of partial abduction and prediction (Alg. 2). This procedure allows for new machinery for decomposing spurious variations in Markovian (Thm. 1) and Semi-Markovian (Thm. 3) models. Finally, we also developed sufficient conditions for identification of such spurious decompositions (Thms. 2, 4), and demonstrated the approach empirically (Sec. 5). The main limitation of our approach is the need for a fully-specified causal diagram, which may be challenging in practice. However, from a fully specified graph and the data, our tools for decomposing spurious effects give a fine-grained quantification of what the main confounders are. As is common in causal inference, the granularity of the obtained knowledge needs to be matched with the strength of the causal assumptions (in this case, specifying the causal diagram). Conversely, in the absence of such assumptions, fine-grained quantitative knowledge about these effects cannot be obtained in general [2], and we hypothesize that precise quantification of spurious effects is not attainable in the absence of a causal diagram. Finally, we discuss another technical solution that may alleviate some of the difficulty of causal modeling. Recently, cluster diagrams have been proposed [1], in which one can consider groups of confounders (instead of considering each confounder separately), and thus the specification of causal assumptions becomes less demanding (due to clustering, the number of nodes in the graph is smaller). However, causal decompositions as described in this paper can still be applied to cluster diagrams. This offers a way to choose a different level of granularity for settings where domain knowledge may not be specific enough to elicit a full causal diagram.

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

**Acknowledgements** This research was supported in part by the NSF, ONR, AFOSR, DoE, Amazon, JP Morgan, and The Alfred P. Sloan Foundation. We would like to thank Inwoo Hwang for providing very useful comments on the paper during the NeurIPS conference.

# Supplementary Material for *A Causal Framework for Decomposing Spurious Variations*

The source code for reproducing the experiments can be found in our code repository.

## A    Theorem and Proposition Proofs

### A.1    Proof of Prop. 2

*Proof.* Note that TV and TE are defined as:

$$\text{TV}_{x_0,x_1}(y) = P(y \mid x_1) - P(y \mid x_0) \tag{32}$$

$$\text{TE}_{x_0,x_1}(y) = P(y_{x_1}) - P(y_{x_0}). \tag{33}$$

We can expand the TV measure in the following way:

$$\text{TV}_{x_0,x_1}(y) = P(y \mid x_1) - P(y \mid x_0) \tag{34}$$

$$= P(y \mid x_1) - P(y_{x_1}) + P(y_{x_1}) - P(y \mid x_0) \tag{35}$$

$$= \underbrace{P(y \mid x_1) - P(y_{x_1})}_{\text{Exp-SE}_{x_1}(y)} + \underbrace{P(y_{x_1}) - P(y_{x_0})}_{\text{TE}_{x_0,x_1}(y)} + \underbrace{P(y_{x_0}) - P(y \mid x_0)}_{-\text{Exp-SE}_{x_0}(y)} \tag{36}$$

$$= \text{TE}_{x_0,x_1}(y) + \text{Exp-SE}_{x_1}(y) - \text{Exp-SE}_{x_0}(y), \tag{37}$$

showing the required result. □

### A.2    Proof of Thm. 1

*Proof.* Note that

$$\sum_{i=0}^{k-1} \text{Exp-SE}_x^{U_{[i]}, U_{[i+1]}}(y) = \sum_{i=0}^{k-1} P(y \mid x^{U_{[i]}}) - P(y \mid x^{U_{[i+1]}}) \tag{38}$$

is a telescoping sum, and thus we have that

$$\sum_{i=0}^{k-1} \text{Exp-SE}_x^{U_{[i]}, U_{[i+1]}}(y) = \sum_{i=0}^{k-1} P(y \mid x^{U_{[i]}}) - P(y \mid x^{U_{[i+1]}}) \tag{39}$$

$$= P(y \mid x^{\emptyset}) - P(y \mid x^{U_{[k]}}) \tag{40}$$

$$= P(y \mid x) - P(y_x) \tag{41}$$

$$= \text{Exp-SE}_x(y), \tag{42}$$

completing the proof of the theorem. □

### A.3    Proof of Thm. 2

*Proof.* Notice that fixing a specific value for the variables $(U_1, \dots, U_k) = (u_1, \dots, u_k)$ also gives a unique value for the variables $(Z_1, \dots, Z_k) = (z_1, \dots, z_k)$. Therefore, we can write

$$P(y \mid x^{U_{[i]}}) = \sum_{u_{[i]}} P(u_{[i]}) P(y \mid x, u_{[i]}) \tag{43}$$

$$= \sum_{u_{[i]}} P(u_{[i]}) P(y \mid x, u_{[i]}, z_{[i]}(u_{[i]})) \tag{44}$$

$$= \sum_{z_{[i]}} \sum_{u_{[i]}} P(u_{[i]}) \mathbb{1}(Z_{[i]}(u_{[i]}) = z_{[i]}) P(y \mid x, z_{[i]}) \tag{45}$$

$$= \sum_{z_{[i]}} P(z_{[i]}) P(y \mid x, z_{[i]}) \tag{46}$$

$$= \sum_{z} P(y \mid x, z) P(z_{-[i]} \mid x, z_{[i]}) P(z_{[i]}). \tag{47}$$

□

The above proof makes use of the fact that the exogenous variables $U_i$ are considered in the topological ordering in the decomposition in Eq. 22, since in this case a fixed value of $u_{[i]}$ implies a fixed value of $z_{[i]}$. However, when considering decompositions that do not follow a topological ordering, this is not the case, and we lose the identifiability property of the corresponding effects, as shown in the example in Appendix B.2.

## A.4 Proof of Thm. 3

*Proof.* The proof is analogous to the proof of Thm. 1, the only difference being that there is no longer a 1-to-1 of the latent variables $U_i$ with the observed confounders $Z_i$. Rather, each $U_i$ may correspond to one or more $Z_i$ variables. However, we still have that

$$\sum_{i=0}^{k-1} \text{Exp-SE}_x^{U_{[i]}, U_{[i+1]}}(y) = \sum_{i=0}^{k-1} P(y \mid x^{U_{[i]}}) - P(y \mid x^{U_{[i+1]}}) \tag{48}$$

is a telescoping sum, and thus we have that

$$\sum_{i=0}^{k-1} \text{Exp-SE}_x^{U_{[i]}, U_{[i+1]}}(y) = \sum_{i=0}^{k-1} P(y \mid x^{U_{[i]}}) - P(y \mid x^{U_{[i+1]}}) \tag{49}$$

$$= P(y \mid x^{\emptyset}) - P(y \mid x^{U_{[k]}}) \tag{50}$$

$$= P(y \mid x) - P(y_x) \tag{51}$$

$$= \text{Exp-SE}_x(y), \tag{52}$$

completing the proof of the theorem. $\qquad \square$

## A.5 Proof of Thm. 4

*Proof.* Let $U_{PA}$ be the set of exogenous variables not updated according to evidence, and suppose that (i) $X, Y \notin \text{AS}(U_{PA})$; (ii) $U_{PA} = \text{an}^{\text{ex}}(AS(U_{PA}))$. Note that

$$P(y \mid x^{U_{PA}}) \overset{\text{(def)}}{=} \sum_{u_{PA}} P(u_{PA}) P(y \mid x, u_{PA}) \tag{53}$$

$$= \sum_{u_{PA}, z_{AS}} P(u_{PA}) P(y \mid x, u_{PA}, z_{AS}) P(z_{AS} \mid x, u_{PA}), \tag{54}$$

where $Z_{AS}$ is the anchor set of $U_{PA}$ and the second line follows from the law of total probability. Consider any exogenous ancestor of $Z_{AS}$, denoted by $U_z$. By condition (ii) of ancestral closure, $U_z$ must be in $U_{PA}$. Therefore, $U_{PA}$ contains all exogenous ancestors of $Z_{AS}$. Consequently, a fixed value of $u_{PA}$ also implies a value of $Z_{AS}$, labeled $z_{AS}$. This means that

$$P(z_{AS} \mid x, u_{PA}) = \mathbb{1}(Z_{AS}(u_{PA}) = z_{AS}). \tag{55}$$

Next, suppose there is an open path from $U_{PA}$ to $Y$ when conditioning on $X, Z_{AS}$, labeled $U_{PA,i} \to Z_s \overset{\to}{\leftarrow} Z_{s'} \to \cdots \to Y$. By definition of the anchor set, $Z_{AS}$ must contain the first variable on this path, $Z_s$, and $Z_s$ is different from $X, Y$. Consider first the case with the arrow from $Z_s$ outgoing, $U_{PA,i} \to Z_s \to \cdots \to Y$. When conditioning on $Z_{AS}$, this path is closed since $Z_s \in Z_{AS}$, yielding a contradiction. Consider then the second case with the arrow incoming into $Z_s$, $U_{PA,i} \to Z_s \leftarrow Z_{s'} \to \cdots \to Y$. Since $Z_{s'}$ points to $Z_s$, $Z_{s'}$ differs from $X, Y$. Furthermore, by anchor set exogenous ancestral closure, the exogenous variable of $Z_{s'}$, labeled $U_{s'}$, must also be in $U_{PA}$. Hence, $Z_{AS}$ contains $Z_{s'}$, and $Z_{s'}$ cannot be a collider on this path, so conditioning on $Z_{AS}$ blocks the path, again yielding a contradiction. We conclude that no open path between $U_{PA}$ and $Y$ exists when conditioning on $Z_{AS}, X$. Therefore, it holds that

$$P(y \mid x, u_{PA}, z_{AS}) = P(y \mid x, z_{AS}). \tag{56}$$

Finally, by plugging in Eqs. 55-56 into Eq. 54 we obtain that

$$P(y \mid x^{U_{PA}}) = \sum_{u_{PA}, z_{AS}} P(u_{PA}) P(y \mid x, z_{AS}) \mathbb{1}(Z_{AS}(u_{PA}) = z_{AS}) \tag{57}$$

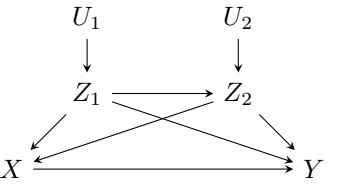

Figure 7: Markovian causal diagram used in Ex. 4 with explicitly drawn latent variables $U_1, U_2$.

$$= \sum_{z_{AS}} P(y \mid x, z_{AS}) \underbrace{\sum_{u_{PA}} P(u_{PA}) \mathbb{1}(Z_{AS}(u_{PA}) = z_{AS})}_{P(z_{AS}) \text{ by definition}} \tag{58}$$

$$= \sum_{z_{AS}} P(y \mid x, z_{AS}) P(z_{AS}), \tag{59}$$

therefore witnessing identifiability of $P(y \mid x^{U_{PA}})$ and completing the proof. $\qquad\square$

# B  Examples

## B.1  Markovian Decomposition Example

**Example 4** (Latent variable attribution in a Markovian model)**.** *Consider the following SCM $\mathcal{M}^*$:*

$$\mathcal{M}^* : \begin{cases} Z_1 \leftarrow B(0.5) & (60) \\ Z_2 \leftarrow B(0.4 + 0.2Z_1) & (61) \\ X \leftarrow B(0.3 + 0.2Z_1 + 0.2Z_2) & (62) \\ Y \leftarrow X + Z_1 + Z_2, & (63) \end{cases}$$

*and the causal diagram in Fig. 7. We wish to decompose the quantity Exp-SE$_x(y)$ into the variations attributed to the latent variables $U_1, U_2$. Following the decomposition from Thm. 1 we can write*

$$\textit{Exp-SE}_x(y \mid x_1) = \underbrace{\mathbb{E}(y \mid x_1) - \mathbb{E}(y \mid x_1^{U_1})}_{U_1 \text{ contribution}} \tag{64}$$
$$+ \underbrace{\mathbb{E}(y \mid x_1^{U_1}) - \mathbb{E}(y \mid x_1^{U_1, U_2})}_{U_2 \text{ contribution}}.$$

*We now need to compute the terms appearing in Eq. 64. In particular, we know that*

$$\mathbb{E}(y \mid x_1^{U_1, U_2}) = \mathbb{E}(y \mid do(x_1)) \tag{65}$$
$$= 1 + \mathbb{E}(Z_1 \mid do(x_1)) + \mathbb{E}(Z_2 \mid do(x_1)) \tag{66}$$
$$= 1 + \mathbb{E}(Z_1) + \mathbb{E}(Z_2) = 1 + 0.5 + 0.5 = 2. \tag{67}$$

*Similarly, we can also compute*

$$\mathbb{E}(y \mid x_1) = 1 + P(Z_1 = 1 \mid x_1) + P(Z_2 = 1 \mid x_1), \tag{68}$$

*where $P(Z_1 = 1 \mid x_1)$ can be expanded as*

$$P(Z_1 = 1 \mid x_1) = \frac{P(Z_1 = 1, X = 1)}{P(X = 1)} \tag{69}$$

$$= \frac{P(Z_1 = 1, X = 1, Z_2 = 1) + P(Z_1 = 1, X = 1, Z_2 = 0)}{P(X = 1)} \tag{70}$$

$$= \frac{0.5 * 0.6 * 0.7 + 0.5 * 0.4 * 0.5}{0.5} = 0.62. \tag{71}$$

*The value of $P(Z_2 = 1 \mid x_1)$ is computed analogously and also equals 0.62, implying that $\mathbb{E}(y \mid x_1) = 1 + 0.62 + 0.62 = 2.24$. Finally, we want to compute $\mathbb{E}(y \mid x_1^{U_1})$, which equals*

$$\mathbb{E}(y \mid x_1^{U_1}) = 1 + P(Z_1 = 1 \mid x_1^{U_1}) + P(Z_2 = 1 \mid x_1^{U_1}). \tag{72}$$

(a) Exp-SE$_x^{\emptyset, U_1}(y)$.  (b) Exp-SE$_x^{U_1, \{U_1, U_2\}}(y)$.

Figure 8: Graphical representation of Exp-SE effect decomposition in Ex. 4.

By definition, $P(Z_1 = 1 \mid x_1^{U_1}) = P(Z_1 = 1) = 0.5$. For $P(Z_2 = 1 \mid x_1^{U_1})$ we write

$$P(Z_2 = 1 \mid x_1^{U_1}) = \sum_{z_1} P(Z_2 = 1 \mid x_1, z_1) P(z_1) \tag{73}$$

$$= \frac{1}{2} \left[ \frac{P(Z_2 = 1, X = 1, Z_1 = 1)}{P(X = 1, Z_1 = 1)} + \frac{P(Z_2 = 1, X = 1, Z_1 = 0)}{P(X = 1, Z_1 = 0)} \right] \tag{74}$$

$$= \frac{1}{2} \left[ \frac{0.21}{0.31} + \frac{0.21}{0.31} \right] \approx 0.68, \tag{75}$$

implying that $\mathbb{E}(y \mid x_1^{U_1}) = 2.18$. Putting everything together, we found that

$$\underbrace{\text{Exp-SE}_x(y \mid x_1)}_{=0.24} = \underbrace{\text{Exp-SE}_x^{\emptyset, U_1}(y \mid x_1)}_{=0.06 \text{ from } U_1} + \underbrace{\text{Exp-SE}_x^{U_1, \{U_1, U_2\}}(y \mid x_1)}_{=0.18 \text{ from } U_2}. \tag{76}$$

The terms appearing on the r.h.s. of Eq. 76 are shown as graphical contrasts in Fig. 8. On the left side of Fig. 8a, $U_1, U_2$ are responding to the conditioning $X = x$, compared against the right side where only $U_2$ is responding to the conditioning $X = x$. In the second term, in Fig. 8b, on the left only $U_2$ responds to $X = x$, compared against the right side in which neither $U_1$ nor $U_2$ respond to $X = x$ conditioning.

## B.2 Non-topological Counterexample

**Example 5** (Non-identification of latent spurious decomposition). *Consider two SCMs $\mathcal{M}_1, \mathcal{M}_2$. Both SCMs have the same set of assignment equations $\mathcal{F}$, given by*

$$\mathcal{F} := \begin{cases} Z_1 \leftarrow U_1 & (77) \\[2mm] Z_2 \leftarrow \begin{cases} Z_1 & \text{if } U_2 = 1 \\ 1 - Z_1 & \text{if } U_2 = 2 \\ 1 & \text{if } U_2 = 3 \\ 0 & \text{if } U_2 = 4 \end{cases} & (78) \\[6mm] X \leftarrow (Z_1 \wedge U_{X1}) \vee (Z_2 \wedge U_{X2}) \vee U_X & (79) \\[2mm] Y \leftarrow X + Z_1 + Z_2, & (80) \end{cases}$$

and the causal diagram given in Fig. 7. The two SCMs differ in the distribution over the latent variables. In particular, for $\mathcal{M}_1$ we have

$$P^{\mathcal{M}_1}(U) : \begin{cases} U_1, U_{X1}, U_{X2}, U_X \sim \text{Bernoulli}(0.5) & (81) \\[2mm] U_2 \sim \text{Multinom}(4, 1, (0, \frac{1}{4}, \frac{1}{2}, \frac{1}{4})), & (82) \end{cases}$$

and for $\mathcal{M}_2$

$$P^{\mathcal{M}_2}(U) : \begin{cases} U_1, U_{X1}, U_{X2}, U_X \sim \text{Bernoulli}(0.5) & (83) \\[2mm] U_2 \sim \text{Multinom}(4, 1, (\frac{1}{4}, \frac{1}{2}, \frac{1}{4}, 0)). & (84) \end{cases}$$

That is, the only difference between $P^{\mathcal{M}_1}(U)$ and $P^{\mathcal{M}_2}(U)$ is in how $U_2$ attains its value. In fact, one can check that the observational distributions $P^{\mathcal{M}_1}(V)$ and $P^{\mathcal{M}_2}(V)$ are the same. However,

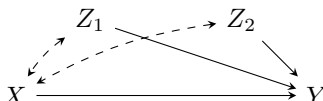

Figure 9: Causal diagram appearing in Exs. 6-7.

*when computing $\mathbb{E}^{\mathcal{M}}(y \mid x_0^{U_2})$ we have that*

$$\mathbb{E}^{\mathcal{M}_1}(y \mid x_0^{U_2}) = 1 \tag{85}$$

$$\mathbb{E}^{\mathcal{M}_2}(y \mid x_0^{U_2}) = 0.93, \tag{86}$$

*showing that the quantity $\mathbb{E}^{\mathcal{M}}(y \mid x_0^{U_2})$ is non-identifiable.*

The example illustrates that even in the Markovian case, when the variables are not considered in a topological order (in the example above, the variable $U_2$ was considered without the variable $U_1$ being added first), we might not be able to identify the decomposition of the spurious effects.

### B.3 Semi-Markovian Decomposition Example

**Example 6** (Semi-Markovian spurious decomposition). *Consider the following SCM $\mathcal{M}$:*

$$\mathcal{F}, P(U) : \begin{cases} Z_1 \leftarrow U_1 \wedge U_{1X} & (87) \\ Z_2 \leftarrow U_2 \vee U_{2X} & (88) \\ X \leftarrow U_X \wedge (U_{1X} \vee U_{2X}) & (89) \\ Y \leftarrow X + Z_1 + Z_2 & (90) \\ \\ U_1, U_2, U_{1X}, U_{2X}, U_X \stackrel{i.i.d.}{\sim} Bernoulli(0.5). & (91) \end{cases}$$

*The causal diagram $\mathcal{G}$ associated with $\mathcal{M}$ is given in Fig. 9. The exogenous variables that lie on top of a spurious trek are $U_{1X}, U_{2X}$. Therefore, following the decomposition from Thm. 3, we can attribute spurious variations to these two variables:*

$$\textit{Exp-SE}_x(y \mid x_1) = \underbrace{\mathbb{E}(y \mid x_1) - \mathbb{E}(y \mid x_1^{U_{1X}})}_{U_{1X} \text{ contribution}} \tag{92}$$

$$+ \underbrace{\mathbb{E}(y \mid x_1^{U_{1X}}) - \mathbb{E}(y \mid x_1^{U_{1X}, U_{2X}})}_{U_{2X} \text{ contribution}}.$$

*We now compute the terms appearing in Eq. 92. In particular, we know that*

$$\mathbb{E}(y \mid x_1^{U_{1X}, U_{2X}}) = \mathbb{E}(y \mid do(x_1)) = 1 + \mathbb{E}(Z_1 \mid do(x_1)) + \mathbb{E}(Z_1 \mid do(x_1)) \tag{93}$$

$$= 1 + \mathbb{E}(Z_1) + \mathbb{E}(Z_2) = 1 + 0.25 + 0.75 = 2. \tag{94}$$

*Similarly, we can also compute*

$$\mathbb{E}(y \mid x_1) = 1 + P(Z_1 = 1 \mid x_1) + P(Z_2 = 1 \mid x_1), \tag{95}$$

*Now, $P(Z_1 = 1 \mid x_1) = \frac{P(Z_1=1, x_1)}{P(x_1)}$, and we know that $X = 1$ if and only if $U_X = 1$ and $U_{1X} \vee U_{2X} = 1$, which happen independently with probabilities $\frac{1}{2}$ and $\frac{3}{4}$, respectively. Next, $Z_1 = 1, X = 1$ happens if and only if $U_X = 1, U_{1X} = 1$ and $U_1 = 1$, which happens with probability $\frac{1}{8}$. Therefore, we can compute*

$$P(Z_1 = 1 \mid x_1) = \frac{\frac{1}{8}}{\frac{1}{2} * \frac{3}{4}} = \frac{1}{3}. \tag{96}$$

*Furthermore, we similarly compute that $Z_2 = 1, X = 1$ happens if either $U_X = 1, U_{2X} = 1$ or $U_X = 1, U_{2X} = 0, U_2 = 1, U_{1X} = 1$ which happens disjointly with probabilities $\frac{1}{4}, \frac{1}{16}$, respectively. Therefore,*

$$P(Z_2 = 1 \mid x_1) = \frac{\frac{1}{4} + \frac{1}{16}}{\frac{1}{2} * \frac{3}{4}} = \frac{5}{6}. \tag{97}$$

*Putting everything together we obtain that*

$$\mathbb{E}(y \mid x_1) = 1 + \frac{1}{3} + \frac{5}{6} = \frac{13}{6}. \tag{98}$$

*Finally, we want to compute $\mathbb{E}(y \mid x_1^{U_{1X}})$, which equals*

$$\mathbb{E}(y \mid x_1^{U_{1X}}) = 1 + P(Z_1 = 1 \mid x_1^{U_{1X}}) + P(Z_2 = 1 \mid x_1^{U_{1X}}). \tag{99}$$

*Now, to evaluate these expressions, we distinguish two cases, namely (i) $U_{1X} = 1$ and (ii) $U_{1X} = 0$. In the first case, $P(Z_1 \mid x_1) = \frac{1}{2}$ and $P(Z_2 = 1 \mid x_1) = \frac{3}{4}$. In the second case, $P(Z_1 \mid x_1) = 0$ and $P(Z_2 = 1 \mid x_1) = 1$. Therefore, we can compute*

$$P(Z_1 = 1 \mid x_1^{U_{1X}}) = \frac{1}{2} P_{U_{1X}=1}(Z_1 \mid x_1) + \frac{1}{2} P_{U_{1X}=0}(Z_1 \mid x_1) = \frac{1}{4} \tag{100}$$

$$P(Z_2 = 1 \mid x_1^{U_{1X}}) = \frac{1}{2} P_{U_{1X}=1}(Z_2 \mid x_1) + \frac{1}{2} P_{U_{1X}=0}(Z_2 \mid x_1) = \frac{7}{8}, \tag{101}$$

*which implies that $\mathbb{E}(y \mid x_1^{U_{1X}}) = \frac{17}{8}$. Finally, this implies that*

$$\underbrace{Exp\text{-}SE_x(y \mid x_1)}_{=\frac{1}{6}} = \underbrace{Exp\text{-}SE_x^{\emptyset, U_{1X}}(y \mid x_1)}_{=\frac{1}{24} \text{ from } U_{1X}} + \underbrace{Exp\text{-}SE_x^{U_{1X}, \{U_{1X}, U_{2X}\}}(y \mid x_1)}_{=\frac{1}{8} \text{ from } U_{2X}}. \tag{102}$$

The terms appearing on the r.h.s. of Eq. 102 are shown as graphical contrasts in Fig. 4. On the left side of Fig. 4a, $U_{1X}, U_{2X}$ are responding to the conditioning $X = x$, compared against the right side where only $U_{2X}$ is responding to the conditioning $X = x$. In the second term, in Fig. 4b, on the left only $U_{2X}$ responds to $X = x$, compared against the right side in which neither $U_{1X}$ nor $U_{2X}$ respond to $X = x$ conditioning.

## B.4 Semi Markovian Identification Examples

**Example 7** (Spurious Treks). *Consider the causal diagram in Fig. 7. In the diagram, latent variables $U_1, U_2$ both lie on top of a spurious trek because:*

$$X \leftarrow Z_1 \leftarrow U_1 \rightarrow Z_1 \rightarrow Y \text{ is a spurious trek with top } U_1$$
$$X \leftarrow Z_2 \leftarrow U_2 \rightarrow Z_2 \rightarrow Y \text{ is a spurious trek with top } U_2.$$

*There are also other spurious treks with $U_1$ on top, such as $X \leftarrow Z_1 \leftarrow U_1 \rightarrow Z_1 \rightarrow Z_2 \rightarrow Y$.*

**Example 7** (continued - $U_{sToT}$ construction). *We continue with Ex. 6 and the causal graph in Fig. 9 and perform the steps as follows:*

    *(i) initialize $U_{sToT} = \emptyset$,*

    *(ii) note that $\{X, Z_1\}$ create a maximal clique, since:*

        *(a) they are connected with a bidirected edge and thus form a clique,*

        *(b) $\{X, Z_1, Z_2\}$ do not form a clique, due to the bidirected edge $Z_1 \leftarrow\!-\!\rightarrow Z_2$ not being present,*

        *(c) $\{X, Z_1, Y\}$, $\{X, Z_1, Z_2, Y\}$ do not form a clique, since $Y$ is not incident to any bidirected edges,*

        *(d) thus, the clique $\{X, Z_1\}$ is also maximal.*

        *Let the variable $U_{1X}$ be associated with this clique, pointing to $X, Z_1$, and note that $U_{1X}$ lies on top of a spurious trek between $X, Y$,*

    *(iii) similarly, $\{X, Z_2\}$ also create a maximal clique, associated with the variable $U_{X2}$, pointing to $X, Z_2$, that lies on top of a spurious trek between $X, Y$,*

    *(iv) the node $Y$ also forms a maximal clique and is associated with the variable $U_Y$ that does not lie on top of a spurious trek (it does not have a path to $X$).*

*Therefore, we have constructed the set $U_{sToT} = \{U_{1X}, U_{2X}\}$.*

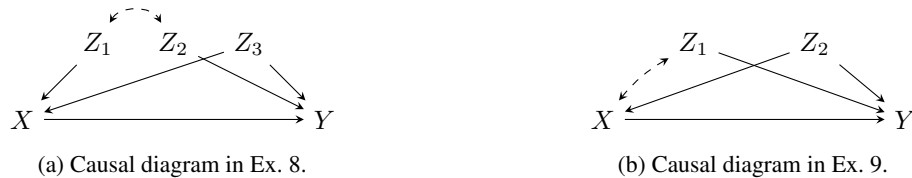

(a) Causal diagram in Ex. 8.          (b) Causal diagram in Ex. 9.

Figure 10: Causal diagrams in Exs. 8-9.

**Example 7** (continued - anchor set). *For the set $U_{sToT} = \{U_{1X}, U_{2X}\}$ associated with the causal diagram in Fig. 9, the anchor sets can be computed as follows:*

$$AS(U_{1X}) = \{X, Z_1\}, \tag{103}$$
$$AS(U_{2X}) = \{X, Z_2\}, \tag{104}$$
$$AS(U_{1X}, U_{2X}) = \{X, Z_1, Z_2\}. \tag{105}$$

**Example 7** (continued - anchor set exogenous ancestral closure). *Consider the causal diagram in Fig. 9. With respect to the diagram, we have that*

$$\text{an}^{ex}(AS(U_{1X})) = \text{an}^{ex}(X, Z_1) = \{U_{1X}, U_{2X}\}, \tag{106}$$
$$\text{an}^{ex}(AS(U_{2X})) = \text{an}^{ex}(X, Z_2) = \{U_{1X}, U_{2X}\}, \tag{107}$$
$$\text{an}^{ex}(AS(\{U_{1X}, U_{2X}\})) = \text{an}^{ex}(X, Z_1, Z_2) = \{U_{1X}, U_{2X}\}. \tag{108}$$

*Therefore, $\{U_{1X}, U_{2X}\}$ satisfies anchor set exogenous ancestral closure, whereas $U_{1X}$ and $U_{2X}$ do not, since for instance $U_{1X}$ has $X$ in its anchor set, but $X$ has $U_{2X}$ as its ancestor.*

We now consider an example of effect identification based on Thm. 4:

**Example 8** (Thm. 4 Application). *Consider the causal diagram in Fig. 10a. Consider the query $\mathbb{E}(y \mid x_1^{U_{12}})$ associated with a partially abducted submodel in which the noise variable $U_{12}$ determining the values of $Z_1, Z_2$ is not updated according to evidence. Based on Thm. 4, we verify that*

*(i) $X, Y$ are not in the anchor set $AS(U_{12}) = \{Z_1, Z_2\}$;*

*(ii) $\text{an}^{ex}(AS(U_{12})) = \text{an}^{ex}(Z_1, Z_2) = U_{12}$ meaning that $U_{12}$ satisfies anchor set exogenous ancestral closure (ASEAC).*

*Therefore, the query $\mathbb{E}(y \mid x_1^{U_{12}})$ is identifiable from observational data. To witness, we expand the query as:*

$$\mathbb{E}(y \mid x_1^{U_{12}}) = \sum_{u_{12}} P(u_{12}) \mathbb{E}(y \mid x_1, u_{12}) \tag{109}$$

$$= \sum_{u_{12}, z_1, z_2} P(u_{12}) \mathbb{E}(y \mid x_1, u_{12}, z_1, z_2) \mathbb{1}(Z_1(u_{12}) = z_1, Z_2(u_{12}) = z_2) \tag{110}$$

$$= \sum_{u_{12}, z_1, z_2} P(u_{12}) \mathbb{E}(y \mid x_1, z_1, z_2) \mathbb{1}(Z_1(u_{12}) = z_1, Z_2(u_{12}) = z_2) \tag{111}$$

$$= \sum_{z_1, z_2} \mathbb{E}(y \mid x_1, z_1, z_2) \sum_{u_{12}} P(u_{12}) \mathbb{1}(Z_1(u_{12}) = z_1, Z_2(u_{12}) = z_2) \tag{112}$$

$$= \sum_{z_1, z_2} \mathbb{E}(y \mid x_1, z_1, z_2) P(z_1, z_2) \tag{113}$$

$$= \sum_{z_1, z_2, z_3} \mathbb{E}(y \mid x_1, z_1, z_2, z_3) P(z_1, z_2) P(z_3 \mid x_1, z_1, z_2), \tag{114}$$

*providing an identification expression from observational data.*

## C  Discussion of Thm. 4

Thm. 4 introduces a sufficient condition for the identification of quantities under partial abduction (Def. 2). Here, we discuss why some of the conditions in the theorem are necessary, and provide an example that further elucidates the theorem's scope.

**Example 9** (Non-Identification in Semi-Markovian Models). *Consider the causal diagram in Fig. 10b and consider two SCMs $\mathcal{M}_1, \mathcal{M}_2$ constructed as follows. Both SCMs have the same set of assignment equations $\mathcal{F}$, given by*

$$\mathcal{F} := \begin{cases} Z_1 \leftarrow \begin{cases} 1 & \text{if } U_{1X} > 4 \\ 0 & \text{if } U_{1X} \leq 4 \end{cases} & (115) \\[2ex] Z_2 \leftarrow U_2 & (116) \\[2ex] X \leftarrow \begin{cases} 1 & \text{if } U_{1X} \in \{1,5\} \\ Z_2 & \text{if } U_{1X} \in \{2,6\} \\ 1 - Z_2 & \text{if } U_{1X} \in \{3,7\} \\ 0 & \text{if } U_{1X} \in \{4,8\} \end{cases} & (117) \\[4ex] Y \leftarrow Z_1 \vee Z_2. & (118) \end{cases}$$

*The two SCMs differ in the distribution over the latent variables. In particular, for $\mathcal{M}_1$ we have*

$$P^{\mathcal{M}_1}(U) : \begin{cases} U_2 \sim \text{Bernoulli}(0.6) & (119) \\[2ex] U_{1X} \sim \text{Multinom}(8, 1, (\frac{1}{8}, \frac{1}{8}, \frac{1}{8}, \frac{1}{8}, \frac{1}{8}, \frac{1}{8}, \frac{1}{8}, \frac{1}{8})), & (120) \end{cases}$$

*and for $\mathcal{M}_2$*

$$P^{\mathcal{M}_2}(U) : \begin{cases} U_2 \sim \text{Bernoulli}(0.6) & (121) \\[2ex] U_{1X} \sim \text{Multinom}(8, 1, (0, \frac{1}{4}, \frac{1}{4}, 0, 0, \frac{1}{4}, \frac{1}{4}, 0)), & (122) \end{cases}$$

*That is, the only difference between $P^{\mathcal{M}_1}(U)$ and $P^{\mathcal{M}_2}(U)$ is in how $U_{1X}$ attains its value. Furthermore, we can verify that $\mathcal{M}_1$ and $\mathcal{M}_2$ generate the same observational distribution, given by following distribution table*

| $Z_1$ | $Z_2$ | $X$ | $P(Z_1, Z_2, X)$ |
|-------|-------|-----|------------------|
| 0 | 0 | 0 | 0.10 |
| 1 | 0 | 0 | 0.10 |
| 0 | 1 | 0 | 0.15 |
| 1 | 1 | 0 | 0.15 |
| 0 | 0 | 1 | 0.10 |
| 1 | 0 | 1 | 0.10 |
| 0 | 1 | 1 | 0.15 |
| 1 | 1 | 1 | 0.15 |

*and $Y$ given simply as a deterministic function of $Z_1, Z_2$. Now, suppose we are interested in computing the conditional probability of $Y$ given $X = x_1$ in a partially abducted submodel where $U_{1X}$ does not respond to evidence. The quantity $\mathbb{E}(y \mid x_1^{U_{1X}})$ can be computed as*

$$\mathbb{E}(y \mid x_1^{U_{1X}}) = \sum_{u_{1x}=1}^{8} P(u_{1x}) \mathbb{E}(y \mid x_1, u_{1x}) \tag{123}$$

$$= \sum_{u_{1x}=1}^{8} P(u_{1x}) \mathbb{E}(y \mid x_1, u_{1x}, z_2) P(z_2 \mid x_1, u_{1x}) \tag{124}$$

$$= \sum_{u_{1x}=1}^{8} P(u_{1x}) P(z_2 \mid x_1, u_{1x}) [Z_1(u_{1x}) \vee z_2], \tag{125}$$

*where $P(Z_2 = 1 \mid x_1, u_{1x}) = 0.6$ for $u_{1x} \in \{1, 4, 5, 8\}$, $P(Z_2 = 1 \mid x_1, u_{1x}) = 0$ for $u_{1x} \in \{3, 7\}$, and $P(Z_2 = 1 \mid x_1, u_{1x}) = 1$ for $u_{1x} \in \{4, 8\}$. Evaluating the expression in Eq. 125 for the two*

*SCMs yields*

$$\mathbb{E}^{\mathcal{M}_1}(y \mid x_1^{U_{1X}}) = \frac{31}{40} \tag{126}$$

$$\mathbb{E}^{\mathcal{M}_2}(y \mid x_1^{U_{1X}}) = \frac{3}{4}, \tag{127}$$

*demonstrating that the quantity $\mathbb{E}(y \mid x_1^{U_{1X}})$ is not identifiable for the diagram in Fig. 10b.*

The above example provides some intuition about why the variable $X$ cannot be in the anchor set of the variables $U_{PA}$ that are not updated according to evidence. Very similarly, an analogous example demonstrating that the variable $Y$ cannot be in the anchor set of $U_{PA}$ can also be constructed.

