# OpenReview forum: "A Causal Framework for Decomposing Spurious Variations"
_NeurIPS.cc/2023/Conference — NeurIPS 2023 poster_

### Official Review · Reviewer_EguC · 2023-06-27

**Soundness:** 4 excellent
**Presentation:** 3 good
**Contribution:** 3 good
**Rating:** 8
**Confidence:** 3

**Summary:**

The paper proposes a novel theoretical framework for decomposing spurious causal effects. For this purpose, the authors introduce the concept of partial abduction and prediction. Here, only a subset of the latent variables in a given structural causal model (SCM) is updated when conditioning on observed evidence. The authors use this concept to derive nonparametric decompositions of spurious effects and provide identifiability results both for Markovian and semi-markovian SCMs. The paper ends with an empirical example using the COMPAS dataset.

**Strengths:**

- The paper addresses a relevant problem: decomposing the spurious effect in the presence of multiple confounders. This has potential applications in e.g., causal algorithmic fairness.

- Novelty: I am not aware of any other work done in this regard. The authors develop a novel theory for defining such decompositions based on Pearl's SCM framework.

- The paper is technically solid and rigorous. Definitions are clear and precise, all assumptions are stated transparently, and proof and illustrative examples for theoretical results are provided.

**Weaknesses:**

- My only main concerns are regarding the applicability of the proposed framework in practice. To my understanding, the full (semi-markovian) causal graph needs to be known (including the spurious arrows between the observed confounders) in order to apply the decomposition result from Theorem 3. This may be the case for simple examples like the COMPAS dataset. However, in many applications, confounders are potentially high-dimensional and the causal graph is unknown.

- I think the paper would benefit from expanding the experiment section, i.e., by including a second example dataset.


Minor points:

- Eq. (7): I suggest using the notation $p(Y_x = y)$ so that $y$ on the right-hand side of the equation is well-defined.
- Typo in Definition 7: path causal path
- Proposition 2: I suggest moving the definitions of TV and TE from Appendix A1 to the main paper.


**Questions:**

- It seems like the decomposition in Theorem 1 depends on the chosen topological ordering of the observed confounders $Z_i$. Consequently, would the "spurious contribution" of confounder $Z_i$ change when selecting a different ordering? This would seem counterintuitive for a framework that aims at decomposing the overall spurious effect into individual "spurious contributions".



**Limitations:**

I suggest including a section on limitations. Here, the authors should particularly stress the assumptions necessary for the identification of spurious decompositions from observational data. Furthermore, a discussion on these assumptions in the context of the empirical example in Section 5 should be included.

---

> ### Author Rebuttal · Authors · 2023-08-09
>
> We thank the reviewer for the time and effort in sharing his/her thoughts and comments. We respond point-by-point to each of the concerns raised.
>
> ---
>
> [W1: Required causal assumptions]  Indeed, you are correct that a fully specified graph is needed for Theorem 3. We also agree that this may be challenging to obtain in practice. However, one also needs to consider that fine-grained spurious (or causal / mediated) effects can be inferred from such a fully specified graph (and data of course). In particular, our tools decompose the spurious effect, and give a fine-grained quantification of what the main confounders are. As is common in causal inference, the granularity of the obtained knowledge needs to be matched with the strength of the causal assumptions. Conversely, in the absence of such assumptions, fine-grained knowledge usually cannot be obtained. For instance, it has been shown in the literature in the result known as the causal hierarchy theorem  [1, Theorem 1], which shows that no claim can be made about higher level counterfactuals from lower level data (observational or experimental) without causal assumption, which we consider through the language of causal graphs.
>
>
> Having said that, there is a possible relaxation of our approach, based on the recent development of cluster diagrams [2], which offers some formal way of trading off knowledge versus quantitative predictions. For cluster DAGs, one can consider groups of confounders, and thus the specification of causal assumptions becomes easier and less demanding. However, causal decompositions can still be applied to cluster diagrams. This provides a way to choose a different level of granularity for settings where domain knowledge may not be specific enough to elicit a full causal diagram.
>
> [1] Bareinboim, Elias, et al. "On Pearl's Hierarchy and the Foundations of Causal Inference." In Probabilistic and Causal Inference: The Works of Judea Pearl, ACM, Special Turing Series, pp. 507-556, 2022.
>
> [2] Anand, Tara V., et al. "Effect identification in cluster causal diagrams." Proceedings of the 37th AAAI Conference on Artificial Intelligence. 2023.
>
> ---
>
> [W2: Adding experiments] Thanks for the suggestion; please see the global response [G1]. We now add experiments with a known ground truth.
>
> ---
>
> [W3: notation $p(Y_x = y)$] The reviewer is indeed correct that this is a slight abuse of notation. However, this notation is standard in the graphical approach to causal inference, which makes us afraid that deviating from it may hurt the legibility to causal readers familiar with it. (For example, this is the notation in place at least since the Causality book, Pearl, 2000.)  Still, to address the concern, we added an explicit remark saying there is a slight abuse of notation, and explaining the $Y = y$ is sometimes just replaced with $y$.
>
> ---
>
> [W4: Minor points] Thanks for all of these. We have fixed all three!
>
> ---
>
> [Q1: Topological Ordering] Thanks for raising this interesting point. In fact, the decomposition may change when the topological ordering is changed. However, this should not come as a major surprise, since the decomposition for path-specific causal effects has exactly the same property (i.e., depends on the ordering). However, in our setting, the topological ordering can also be seen as the "natural" ordering, and it is also the only identifiable one. Additionally, one may also consider that the decomposition is unique (independent of ordering) for the additive case (this is again similar to the path-specific causal effects). Let us know if we can elaborate further on this note.
>
> ---
>
> [L1: Limitations] Thanks, please see global response [G2] which describes the discussion we added. Furthermore, regarding the necessary assumptions for identification, our key proposal is to bring some of thee examples from the appendix into the main text. In particular, in Example 7 (currently Appendix B.4), we ground the definitions of an anchor set, and also anchor set exogenous ancestral closure. We believe that adding this will increase the transparency of the proposed definitions, and make more explicit the types of assumptions that are needed for decomposing spurious effects in the Semi-Markovian setting.

---

> > ### Comment · Reviewer_EguC · 2023-08-10
> > **Thank you for your clarifications**
> >
> > I appreciate the detailed response and additional experiments. I still have some concerns regarding practical applicability, but as pointed out by the authors, strong assumptions are to be expected and are in line with previous literature. Overall the paper is a valuable contribution to the causal inference literature. I raise my score accordingly and recommend acceptance.

---

> > > ### Author Response · Authors · 2023-08-11
> > >
> > > We thank the reviewer for acknowledging our response, and providing swift feedback. We also acknowledge the strength of assumptions needed for using the tool, but remark that we are in particular optimistic about the usage of cluster diagrams in this context.
> > >
> > > In particular, we are grateful that the reviewer sees our framework as a valuable contribution to the causal inference literature, which is quite encouraging. Also, thank you for adjusting the score!

---

### Official Review · Reviewer_1quh · 2023-07-08

**Soundness:** 4 excellent
**Presentation:** 3 good
**Contribution:** 3 good
**Rating:** 7
**Confidence:** 2

**Summary:**

This paper considers the problem of decomposing spurious variations in both Markovian and Semi-Markovian models, which are potentially useful to many real applications. A non-parametric  decomposition of spurious effects is given given and identification issue is also discussed under suitable conditions.

**Strengths:**

- decomposing spurious variations can be potentially useful to many real applications
- Presentation is very good with well-structured preliminary and examples
- Proof is complete and concise

**Weaknesses:**

It seems that there is no discussion on limitation of the developed theory.

**Questions:**

The paper is very clear in both motivation and technical part. As the first work on this problem with a general setting, the content of the present version is sufficient for an acceptance.



**Limitations:**

It seems that there is no discussion on limitations. Developing more identification theories is worth future investigation, as also mentioned by the authors.

---

> ### Author Rebuttal · Authors · 2023-08-09
>
> We thank the reviewer for the time and effort in providing his/her feedback. We were encouraged by the fact that the reviewer appreciated the main strengths of the manuscript. We respond point-by-point to the raised concerns.
>
>
> [W1: Limitations of the developed theory] Thanks for noting this; Please see global response [G2] for a detailed discussion of the limitations, which is now added to the Discussion in the paper!

---

### Official Review · Reviewer_G3e3 · 2023-07-08

**Soundness:** 3 good
**Presentation:** 3 good
**Contribution:** 3 good
**Rating:** 8
**Confidence:** 3

**Summary:**

The authors tackle the problem of decomposing the source of variation in causal analysis.  In many real-world problems, there are many factors that can introduce spurious correlation between a treatment and outcome, and breaking down the contribution of each variable to that spurious correlation can be useful in many fields where an explanation is important.  The authors propose a methodology for both Markovian and semi-Markovian models, based on the 'abduction-action-prediction' method.  It allows for the application of evidence to only a subset of exogenous variables, which then allows us to separate the variation from each exogenous variable.

**Strengths:**

This paper is well-written and well-motivated.  The problem it seeks to address is important, and the introduction does a good job at setting the theoretical foundation and describing the problem.  For the most part, the terminology is introduced at a good pace, making the overall narrative easier to follow. I appreciate the authors interspersing examples with figures, since without them, the paper would likely bog down with all the math.  Overall, I found this a compelling approach to an important problem.

**Weaknesses:**

I think, for such an equation-heavy paper, the notation could be made clearer/more explicit in sections.  For example, Proposition 1 is the first time that we see the $e^{U_1}$ notation, which it seems like means 'this evidence is applied to everything except for $U_1$'.  That's a fairly counter-intuitive notation for such a thing, since generally a subscript or superscript implies that we're using those variables in someway, rather than explicitly excluding them (I realize you need to use them to exclude them, but conceptually, the notation is non-obvious for me).  If I'm not misunderstanding what this notation means, then a more explicit definition before Proposition 1 would be helpful, given how central it is to the rest of the paper.

As another notation point, Theorem 2 uses the notation $z_{-[i]}$, but until now, the $Z_{[i]}$ notation has meant, as defined in Definition 2, "all $Z_i$'s, from 1 to i", and it's not obvious how this translates to a negative.

The COMPAS demonstration is great, but since it clearly lacks ground truth, additional synthetic experiments would be helpful to help demonstrate both the correctness of the theory and how this method can be used in practice on other types of data.

**Questions:**

Is there a list of all of assumptions required for your method?  In Section 4.1, you say that the decomposition needs to follow a topological order of the variables U_i, but are there others?

**Limitations:**

In terms of social impact, this method is more likely to have a positive social impact, since causal inference is often applied in domains where spurious variation is present but where the decisions made can have significant effects on people's lives (e.g., the COMPAS example)

---

> ### Author Rebuttal · Authors · 2023-08-09
>
> We thank the reviewer for the time and effort in sharing his/her thoughts and comments. We were quite glad an encouraged by the fact that the reviewer appreciated the main strengths of our work. We respond point-by-point and hope to have addressed the raised concerns.
>
> ---
>
> [W1: Clearer notation] Thanks for this suggestion; the reviewer’s summary “evidence is applied to everything except for $U_x$” precisely captures the intuition behind the operator (i.e., there is no misunderstanding). However, regarding the choice of superscript / subscript – we have little flexibility here, as we see it. The subscript operator in the causal inference literature is already taken, and it has been used heavily to indicate (and index) interventions. Therefore, the superscript notation is the only option available to us (note that, in general, one may apply different parts of evidence to different latent variables, and thus the indicator for latent variables needs to be specific for each part of the evidence).
>
> Still, we definitely agree on the centrality of such operator, and thus re-write the proposition slightly to make explicit how the operator is defined:
>
> > Let $P(Y = y \mid E = e^{U_1})$ denote the conditional probability of the event $Y = y$ conditional on evidence $E = e$, defined as the probability of $Y = y$ in the PA submodel $\mathcal{M}^{U_1, E=e}$ (i.e., the exogenous variables $U_1$ are not updated according to the evidence). Then, we have that:
> \begin{align}
>     P(Y = y \mid E = e^{U_1}) = \sum_{u_1} P(U_1 = u_1)P(Y = y \mid E = e, U_1 = u_1).
> \end{align}
>
> We hope this addresses your concern but, please, let us know. Additionally, we wonder if the notation
>
> $$ x^{\underline{U_{[i]}}}$$
>
> would be better (the underline indicating the evidence is not updated). We are happy to introduce such a change if the reviewer feels strongly about this (although the reviewer will, just like us, note a slight trade-off between making the notation more bushy vs. more informative).
>
> ---
>
> [W2: $Z_{-[i]}$] Thanks for spotting this glitch, which is an omission on our side and not defined anywhere. We now mention this explicitly in Definition 2. Specifically, the $-[i]$ notation corresponds to the complement, that is:
>
> $$Z_{-[i]} = \{Z_{i+1}, \dots, Z_{k}\}.$$
>
> ---
>
> [W3: Synthetic Experiments] Please, see global response [G1]. We now add synthetic experiments with a known ground truth!
>
> ---
>
> [Q1: Assumptions for the Method] Thanks for asking this, it is indeed a good question. We distinguish three different considerations to try to clarify this point:
>
> (i) Firstly, a fully specified causal diagram is needed for the procedure. This is now also discussed in the limitations paragraph of the Discussion.
>
> (ii) Regarding the topological ordering of variables $U_i$ (which follows from the topological ordering of the $Z_i$) – this part is necessary for the identifiability of the spurious decomposition. However, for defining the decomposition itself, it is not strictly necessary. In other words, there may be a decomposition that does not follow a topological ordering, but we will not be able to compute it from the data. However, the one that does follow a topological ordering will be computable, as shown in Theorem 2. We now add a remark on this in the text!
>
> (iii) A similar consideration is also present in the Semi-Markovian case; here, however, we do not invoke a topological ordering, since there is no one-to-one correspondence with the observed variables. However, the very same intuition is behind the condition called anchor set ancestral closure (Definition 7). That is, for a specific order, as shown in Theorem 4, we will be able to compute the decomposition (otherwise, the decomposition can still be defined, but it cannot be uniquely computed from the data).
>
> We appreciate the thoughtful review and hope these answers help clarify the required assumptions and the provided results. Please let us know if there are any further questions!

---

### Official Review · Reviewer_VtmC · 2023-07-20

**Soundness:** 3 good
**Presentation:** 3 good
**Contribution:** 3 good
**Rating:** 7
**Confidence:** 3

**Summary:**

This paper provides a new framework for decomposing spurious effects and its identification conditions. In addition, they proposed a nonparametric method for decomposing the spurious effect under sufficient conditions.

**Strengths:**

- In general, this paper is well-motivated.
- It provides a procedure to decompose spurious effects.
- The theoretical results in this paper are sound.

**Weaknesses:**

- The paper could clarify the difference between "spurious effect" and "spurious variations", or state if they are used interchangeably. More explanation of how these two terms are defined would help the reader understand if they refer to the same concept.
- More details could be provided on how the procedure for Semi-Markovian spurious decomposition is implemented. In particular, explaining the steps for removing the effects of all exogenous variables would make the approach clearer.
- Comparing the proposed method against existing methods on estimating causal effects could highlight an advantage. Since the proposed method can estimate redundant effects, the resulting causal effect estimates may be more accurate compared to current approaches. This comparison could emphasize the improved performance enabled by accounting for spurious variations.

**Questions:**

 See the weaknesses above.

**Limitations:**

 See the weaknesses above.

---

> ### Author Rebuttal · Authors · 2023-08-09
>
> We thank the reviewer for the time and effort in providing your feedback. We respond to each point in the sequel and would be happy to provide further clarification if you feel suitable.
>
> ---
>
> [W1: Spurious Effect vs. Variation] Thanks; we acknowledge that in the paper the distinction between the two is not entirely clear. We now add the following clarification in the Intro:
>
> “A spurious effect is a quantification of spurious variations (or a subset thereof)”.
>
> Thus, for a spurious effect, we usually have some kind of quantity in mind, for example, the experimental spurious effect (Exp-SE) in the l.h.s. of Eq. (2). Spurious variations are a more broad concept and represent co-variations induced by the latent variables in the SCM, for example, a specific value of a latent $U_z$ that affects both $X$ and $Y$ and induces a correlation between them. Hence, an effect is usually tied to a quantity, whereas variations are a more broad term. In the text, in most places, these two notions could indeed be used interchangeably (although we do not think of them as identical, as described above).  Please let us know if this helps clarify the distinction.
>
> ---
>
> [W2: Details of Semi-Markovian procedure in practice] Thanks for mentioning this. Indeed, some of the examples related to Theorem 4 have been pushed into the appendix due to limitations of space. Our key proposal here is to bring some of these examples into the main text. In particular, in Example 7 (currently Appendix B.4), we ground the definitions of an anchor set, and also anchor set exogenous ancestral closure. We believe that adding this will increase the transparency of the proposed definitions, and make more explicit the types of assumptions that are needed for decomposing spurious effects in the Semi-Markovian setting. We thank you for this suggestion!
>
> ---
>
> [W3: Comparing against existing methods] Thanks for bringing up this point. In fact, for estimating the Exp-SE, one needs to estimate $P(y \mid x)$ (which is easy for a binary $x$), and also $P(y_x)$, which is a causal effect. Therefore, the approach in the paper builds on methods for estimating causal effects, and is not really a competitor in this sense. It would be indeed nice to have a result saying that a decomposition of the Exp-SE quantity can improve the estimation of causal effects. For concreteness, Pearl was able to show that in a special case, a causal (not spurious) effect can be estimated in pieces and then composed so as to obtain a more efficient estimator of this effect [1] ([2] may also be interesting in this context). However, there are challenges of how to do this regarding spurious effects and under more general causal diagrams, which we hope to investigate further in the future.  Again, here our goal is mostly to introduce spurious effects that were not even defined, and also to understand the decomposability and other key properties of this quantity. The natural subsequent question after having this foundational step solidified is to investigate estimation properties, both from a statistical and computational perspective.
>
> [1]  J. Pearl, "What is Gained from Past Learning" UCLA Cognitive Systems Laboratory, Technical Report (R-472), March 2018. Journal of Causal Inference, Causal, Casual, and Curious Section, 6(1), Article 20180005, March 2018. https://doi.org/10.1515/jci-2018-0005
>
> [2] J. Hahn and J. Pearl "Precision of Composite Estimators" UCLA Cognitive Systems Laboratory, Technical Report (R-387), September 2011. Working paper.

---

> > ### Comment · Reviewer_VtmC · 2023-08-13
> > **Thank you for the clarifications**
> >
> > Thank you for the clarifications. I’m happy to increase my evaluation score.

---

> > > ### Author Response · Authors · 2023-08-14
> > >
> > > We thank the reviewer for a constructive review process and acknowledging our clarifications. We are quite encouraged by the grade increase from the reviewer, thanks!

---

### Official Review · Reviewer_dHJ7 · 2023-07-25

**Soundness:** 3 good
**Presentation:** 2 fair
**Contribution:** 2 fair
**Rating:** 5
**Confidence:** 2

**Summary:**

The manuscript studies the decomposition of spurious variations. Specifically, a tool is developed for the non-parametric decompositions in both Markovian and Semi-Markovian models.

**Strengths:**

1. The considered problem is practical. As stated in the introduction, the related tools are almost entirely missing in the literature (though I'm not totally sure about it).

2. The theoretical framework is rather complete.

**Weaknesses:**

1. The empirical evaluation seems to be not strong enough. The proposed tool has only been applied in one specific case. The generality and stableness of the method could be better illustrated by more types of evaluation, perhaps including some simulations.

2. Since causal inference in the presence of latent confounders is a well-studied problem, it will be helpful if more discussion on the considered task and the other related ones can be conducted. This can provide additional insight and broaden the understanding of the subject matter.

**Questions:**

1. What could be the limitations of the proposed tool?

**Limitations:**

I didn't find a discussion on the limitations.

---

> ### Author Rebuttal · Authors · 2023-08-09
>
> We thank the reviewer for the time and effort in providing the comments. We respond point-by-point to the concerns raised. In the strengths, the reviewer states we are solving a practical and important problem, and provide a complete theoretical framework. We kindly ask the reviewer to reconsider the gap between the current literature and the tools proposed in our paper, based on the clarifications provided below.
>
> ---
>
> [W1: Empirical Evaluation] Thanks for the suggestion, and for giving us a chance to clarify this further. We would also like to emphasize the generality of the theoretical framework proposed. We handle both the Markovian and the Semi-Markovian cases in the non-parametric setting. Furthermore, we now also added two synthetic experiments, to test the method on a setting where the ground truth is known. Please also see global response [G1].
>
> ---
>
> [W2: More discussion on considered task] We would like to clarify the issue. While there are indeed methods in the literature for handling latent confounders, the estimation target is almost invariably a causal effect or its conditional / mediated versions. That is, no method addresses the specific challenge investigated in this manuscript, that is, how to decompose spurious variations. Notwithstanding, the problem of understanding spurious variations is quite pervasive in applied sciences, such as epidemiology, and in other contexts such as explainable/fair AI. For instance, exposure to a harmful toxin and the development of a disease may be confounded by other work-related hazards. The newly proposed method allows one to quantify which hazards confound this relationship, and how strong they are, which may be quite important for epidemiologists.
>
> The types of questions our manuscript addresses have not been explicitly considered before. However, the type of reasoning about the strength of the confounders has mostly been done for linear models (where confounding can be associated with the estimated coefficients) or through variable importance. Therefore, our work can be seen as providing the first, non-parametric generalization of these basic types of reasoning, which decomposes the spurious effect quantities. In this sense, we believe it fills in an important gap, complementing the existing literature on the estimation of causal effects, and providing another tool for more precise and fine-grained causal analyses.
>
> ---
>
> [Q1: Limitations] Please see global response [G2]. In particular, the discussion on cluster DAGs shows how some of the causal assumptions may be relaxed.

---

> > ### Comment · Reviewer_dHJ7 · 2023-08-18
> >
> > Thank the authors for the clarification. I will keep my score.

---

### Author Rebuttal · Authors · 2023-08-09

The authors would like to sincerely thank all the reviewers for this paper. The main strengths and novelty were clearly appreciated, and the questions raised were quite useful for us to revise and improve the paper.

In our response, we index all weaknesses with W, questions Q, and limitations L. We do not fully cite reviewer's questions (due to character limit), but try our best to provide a caption for each W/Q/L.

Here, we would like to provide two global responses, which are then cited in the individual responses as well.

---

[G1: Extending the Evaluation] As several reviewers suggested, synthetic experiments where the ground truth is known could be a useful addition. We have conducted such experiments, and we provide an example in the pdf that accompanies our review response (an additional experiment for the Semi-Markovian case will also be added).

Synthetic A experiment explanation (see pdf): For this example, we set the parameters $\lambda_1 = \lambda_2 = \lambda_3$ in the described SCM. We then vary each parameter $\lambda_i \in [0, 0.2]$. This changes the value of the effect associated with latent variable $U_i$. We compute the ground truth values for the spurious effects based on the true SCM, using rejection sampling. We also estimate the values using the estimator described in Theorem 2. The plot in the accompanying pdf shows the result.

In particular, we see that the effects associated with each $U_i$ change as the $\lambda_i$ increases. Furthermore, we see that the associated estimates for the effect are correct, and the ground truth values fall within the 95% confidence interval (CI). Therefore, this confirms the correctness of our approach.

---

[G2: Discussion of the Limitations] The following discussion is added to the main text, to clarify the possible limitations of the proposed approach
> “The main limitation of the approach proposed in the paper is the need for a fully-specified causal diagram, including all the bidirected edges that indicate confounding. Specifying such a graph may be challenging in practice. However, one also needs to consider how much knowledge can be ascertained from a fully specified graph and the data. In particular, our tools decompose the spurious effect, and give a fine-grained quantification of what the main confounders are. Such knowledge can be quite powerful and informative in practice. As is common in causal inference, the granularity of the obtained knowledge needs to be matched with the strength of the causal assumptions (in this case, specifying the causal diagram). Conversely, in the absence of such assumptions, fine-grained quantitative knowledge about these effects cannot be obtained in general. For instance, the causal hierarchy theorem  [1, Theorem 1] shows that claims about counterfactuals from observational or interventional data cannot be made without causal assumption, as usually specified through causal diagrams. We hypothesize a similar result, that is, a precise quantification of spurious effects is not attainable in the absence of a causal diagram.

>Furthermore, another technical solution is possible that may alleviate some of the difficulty of causal modeling. Recently, cluster diagrams have been proposed [2], in which one can consider groups of confounders (instead of considering each confounder separately), and thus the specification of causal assumptions becomes easier and less demanding (i.e., due to clustering, the number of nodes in the graph is smaller). However, causal decompositions as described in this paper can still be applied to cluster diagrams. This offers a way to choose a different level of granularity for settings where domain knowledge may not be specific enough to elicit a full causal diagram, while a causal analysis still needs to be undertaken.

>Finally, we also mention that the identification criteria for Semi-Markovian spurious decompositions provided in Theorem 4 are not complete (i.e., they are sufficient but not necessary). We hope to address this question in future work, and provide an even stronger identification criterion, or prove the completeness of the current one.“

---

### Decision · Program_Chairs · 2023-09-21

**Decision:**

Accept (poster)

**Comment:**

The paper presents a novel problem formulation for understanding spurious variations and has some useful theoretical results to that end. I agree with the reviewers that the problem is quite important, and while there are many empirical papers, this paper makes a unique contribution in studying the types of spurious variations formally.
I encourage the authors to incorporate the reviewers' feedback in their final version, especially a discussion on applicability in practice and limitations.